# Mesenchymal Stem Cell-Derived Exosomes Modulate Angiogenesis in Gastric Cancer

**DOI:** 10.3390/biomedicines11041031

**Published:** 2023-03-27

**Authors:** Fawzy Akad, Veronica Mocanu, Sorin Nicolae Peiu, Viorel Scripcariu, Bogdan Filip, Daniel Timofte, Florin Zugun-Eloae, Magdalena Cuciureanu, Monica Hancianu, Teodor Oboroceanu, Laura Condur, Radu Florin Popa

**Affiliations:** 1Department of Morpho-Functional Sciences II (Pathophysiology), “Grigore T. Popa” University of Medicine and Pharmacy, 16, Universitatii Street, 700115 Iasi, Romania; fawzy-h-akad@d.umfiasi.ro (F.A.);; 2Center for Obesity BioBehavioral Experimental Research, 16, Universitatii Street, 700115 Iasi, Romania; 3Department of Vascular Surgery, “Grigore T. Popa” University of Medicine and Pharmacy, 16, Universitatii Street, 700115 Iasi, Romania; 4Department of Surgery, “Grigore T. Popa” University of Medicine and Pharmacy, 16, Universitatii Street, 700115 Iasi, Romania; 5Department of Morpho-Functional Sciences I (Immunology), “Grigore T. Popa” University of Medicine and Pharmacy, 16, Universitatii Street, 700115 Iasi, Romania; 6Department of Morpho-Functional Sciences II (Pharmacology, Clinical Pharmacology and Algeziology), “Grigore T. Popa” University of Medicine and Pharmacy, 16, Universitatii Street, 700115 Iasi, Romania; 7Department of Pharmacognosy, “Grigore T. Popa” University of Medicine and Pharmacy, 16, Universitatii Street, 700115 Iasi, Romania; 8Department of Family Medicine, Faculty of Medicine, Ovidius University, 900527 Constanta, Romania

**Keywords:** gastric cancer, angiogenesis, neovascularization, tumor microenvironment, mesenchymal stem cells, exosomes

## Abstract

Individualized gastric cancer (GC) treatment aims at providing targeted therapies that translate the latest research into improved management strategies. Extracellular vesicle microRNAs have been proposed as biomarkers for GC prognosis. *Helicobacter pylori* infection influences the therapeutic response to and the drivers of malignant changes in chronic gastritis. The successful use of transplanted mesenchymal stem cells (MSCs) for gastric ulcer healing has raised interest in studying their effects on tumor neovascularization and in potential antiangiogenic therapies that could use mesenchymal stem cell secretion into extracellular vesicles—such as exosomes—in GC cells. The use of MSCs isolated from bone marrow in order to achieve angiogenic modulation in the tumor microenvironment could exploit the inherent migration of MSCs into GC tissues. Bone marrow-derived MSCs naturally present in the stomach have been reported to carry a malignancy risk, but their effect in GC is still being researched. The pro- and antiangiogenic effects of MSCs derived from various sources complement their role in immune regulation and tissue regeneration and provide further understanding into the heterogeneous biology of GC, the aberrant morphology of tumor vasculature and the mechanisms of resistance to antiangiogenic drugs.

## 1. Introduction

Gastric cancer (GC) is highly heterogeneous and its molecular classification is based on the location, histology and immunohistochemical evaluation of human epidermal growth factor receptor 2 (HER2) expression and the replication error (RER) phenotype of tumors and their molecular and genetic profiles. Blood markers proposed for the detection of patients at risk of GC or those likely to benefit from specific treatments include pepsinogen I, ghrelin and various microRNAs [1]. DNA methylation, histone modifications and non-coding RNA molecules (microRNAs and long non-coding RNAs) regulate gene expression in gastric tumors [2]. MicroRNAs (miRNAs) produced by exosomes in the tumor microenvironment have been proposed as biomarkers for the early diagnosis of and for disease progression in GC, as they regulate cell differentiation, proliferation and apoptosis. The prognostic role of miRNA-34a expression in the serum or tissues of GC patients is supported by a negative relationship between miRNA-34a and GC progression, with significantly lower levels in GC tissues and metastases [3]. Decreased miR-539 miRNA in GC tissues is associated with poor 5-year survival [4]. On the other hand, miR-17-5p miRNA is upregulated in surgically retrieved GC tissues and in the plasma of GC patients, and it increases the proliferation of GC cells in vitro by inhibiting GC cell apoptosis and by downregulating the expresion of genes that act as tumor suppressors [5]. Exosomal miR-155 derived from GC cells has been found to suppress adipogenesis in adipose (AD)-MSCs, leading to cancer-associated cachexia [6].

MiRNAs are widespread in mesenchymal stem cells (MSCs). The exosomes secreted by MSCs (MSC-exos) isolated from bone marrow (BM), adipose tissue (AD), dental pulp (DP), umbilical cords (UCs), amniotic fluid (AF) or the placenta could be used as miRNA delivery systems in malignancies [7].

MSC-exos have either a pro- or antiangiogenic effect on tumors through various mechanisms: the ERK1/2 pathway, AKT (protein kinase B)/eNOS pathway or miRNA transport [8]. Exome sequencing studies have provided genetic findings important to the understanding of the development of atrophic gastritis and gastric intestinal metaplasia into cancer and GC resistance to chemotherapeutic agents.

Deregulated angiogenesis promotes gastric tumor growth through the expansion of capillaries that provide nutrients and oxygen to the tumor microenvironment. Tumor growth is supported by the transition from an avascular dormant phase to a vascular expanding one. The leaky tumor endothelial cells of the tumor microenvironment increase the interstitial pressure and interact with various growth factors. Some of these growth factors (vascular endothelial growth factors—VEGFs) are angiogenesis mediators found in hypoxic, acidic areas, created by fast proliferating cancer cells due to their high dividing rate [9,10,11]. GC patients with a high expression of VEGF-A and VEGF-C have been shown to have a worse prognosis, with larger tumor sizes and increased metastatic activity [12].

The goal of genome-guided personalized therapy in gastric cancer has required a study of the relationship between MSC-exos and angiogenesis. A literature search was performed on Pubmed/Medline, Google Scholar and Embase with the keywords “(mesenchymal stem cells OR exosomes OR mesenchymal stem cell-derived exosomes OR MSC-exos) AND (angiogenesis OR angiogenic OR neovascularization OR vascular endothelial growth factors) AND (gastric cancer OR stomach neoplasm)”.

Anti-angiogenic agents—such as apatinib, ramucirumab and bevacizumab—have been so far used in GC treatment, but the results have not been as expected [13]. MSCs support angiogenesis and yet the functions of transplanted stem cells are affected by hypoxia and inflammation in the receiving tissues. MSCs lead to a significant increase in VEGF production and MSCs can be used as an important source of VEGFs [14].

## 2. MSCs in GC

The gastric epithelium renews itself incessantly with the help of adult stem cells, derived either from the base or the neck of antral glands or from the isthmus. Similarly to the gut epithelium, two gastric stem cell populations with a different plasticity, longevity and cycling rate have been hypothesized [15]. MSCs play various anti-inflammatory and immunoregulatory roles, with a regenerative potential based on secretion and modulation rather than the direct replacement of tissues [16]. 

The presence of MSC-like cells has been confirmed in human GC tissues and they share similar characteristics to BM-MSCs [17], but have a different phenotype and function [18]. BM-MSCs, which are usually found in inflammed tissues, have been previously linked to GC development in *Helicobacter felis*-infected mice [19]. *Helicobacter pylori* infection increases MSC proliferation and migration and activates the PI3K-AKT signaling pathway, due to the reduction of glutamine and the metabolite alpha-ketoglutarate (α-kg), produced by the gamma-glutamyltransferase enzyme secreted by *Helicobacter pylori*—as shown by an in vivo study on nude mice [20]. BM-MSCs are considered to be the precursors of GC-MSCs. BM-MSCs show a GC-MSC-like phenotype and function through NF-κB activation following knockdown of miR-155-5p, which is downregulated in GC-MSCs [18]. BM-MSCs promote GC angiogenesis by releasing VEGFs, fibroblast-derived growth factors, platelet-derived growth factors, stromal cell derived factors-1 (SDF-1s) and cytokines [21]. 

MSC-like cells are also found in adjacent non-cancerous gastric tissues, sharing the same fibroblast-like appearance as normal BM-MSCs and being different in respect to their gene profiles [22]. 

GC-MSCs have a higher inflammatory response, with increased levels of IL-6, MCP-1 and VEGF [18]. GC-MSCs show an osteoblastic differentiation potential in vitro [23]. GC-MSCs are also characterized by high levels of fibroblast proteins, α-smooth muscle actin (α-SMA) and vimentin [24]. GC-MSCs create an immunosuppressive tumor microenvironment by impairing the anti-tumor immunity mechanisms mediated by peripheral blood mononuclear cells through Treg/Th17 imbalance, with increased levels of Treg cells and decreased levels of Th17 cells [25].

The effect of GC-MSCs on GC progression, migration and angiogenesis is superior to those of adjacent non-cancerous tissue-derived MSCs (GCN-MSCs) and BM-MSCs in vitro. A list of the pro-angiogenic effects of GC-MSCs is provided in Table 1. 

GC-MSCs have the highest levels of the pro-angiogenic factors VEGF, MIP-2, transforming growth factor TGF-β1, IL-6, and—especially—IL-8, showing the ability of GC-MSCs to enhance GC angiogenesis—mostly by secreting the inflammatory cytokine IL-8. GC cells cultured with GC-MSCs achieve a highly branched structure, suggesting an increased capacity to form tube-like structures [23]. MSC-derived IL-6 induces endothelin-1 (ET-1) secretion from cancer cells, which activates the AKT and ERK pathways in endothelial cells to promote endothelial cell recruitment and tumor neovascularization in gastrointestinal cancer cells. In tumors with diameters higher than 7 mm, angiogenesis plays an important role in tumor growth. MSCs increase tumor angiogenesis within the first 14 days—the earliest stage of tumor development [26]. The trans-differentiation of GC-MSCs into endothelial cells also modifies the entire GC vascular network [27].

GC angiogenesis is stimulated by IL-8 cytokines secreted by GC-MSCs and by the interaction of GC-MSCs with neutrophils that produce pro-angiogenic factors [28]. IL-8 has been shown to enhance the angiogenic effects of human BM-MSCs by stimulating VEGF production through PI3K/AKT and mitogen-activated protein kinase (MAPK)/ERK signaling pathways in cancer cells [14]. The interaction between GC-MSCs and GC-infiltrating neutrophils promotes angiogenesis. The pro-angiogenic activity of neutrophils stimulated with GC-MSCs in a conditioned medium obtained from cultured GC cells has been confirmed by the increased formation of tube-like structures as compared with control neutrophils, as shown by an in vitro endothelial cell tube formation assay performed on human umbilical vein endothelial cells (HUVECs) [29]. GC-MSC-derived angiogenesis can be suppressed by inhibiting the NF-κB/VEGF signaling pathways, which are responsible for the promotion of angiogenesis and the stimulation of VEGF expression in vitro and in vivo in the tumor environment. HUVECs showed decreased tube-formation ability when exposed to GC-MSCs treated with NF-κB inhibitors or a VEGF-neutralization antibody [24].

The miRNAs expressed in GC tissue are not unique to a specific cell type, as tumor stroma contains various mesenchymal cell types. MiRNAs packaged into GC-MSC-exos are delivered to GC cells and mediate GC progression. GC-MSCs promote GC progression by transporting these exosomal miRNAs to GC cells. Aberrant miRNA expression found in miR-214, miR-221 and miR-222 was found to be significantly higher in GC-MSCs and in GC tissues as compared to non-GC-MSCs and non-GC tissues. These increased miRNAs help GC-MSCs to promote lymph node metastasis, venous invasion and tumor node metastasis (TNM) staging in vitro and in vivo. GC-MSCs alter miR-221 expression in GC cells by delivering exosomal miRNAs to target cells [30,31].

MiR-15b-3p packed into exosomes (exo-miR-15b-3p) secreted by BGC-823 cells is increased in GC cells, tissues and serum and promote the malignant transformation of normal gastric mucosa epithelium cells GES-1 by regulating the dynein light chain Tctex-type 1 (DYNLT1)/Caspase-3/Caspase-9 signaling pathway, with decreased expression of the DYNLT1 gene [32].
biomedicines-11-01031-t001_Table 1Table 1Mesenchymal stem cells derived from gastric cancer tissues promote tumor angiogenesis.Pro-Angiogenic Factors Secreted by GC-MSCsPro-Angiogenic Signaling Pathways Activated by GC-MSCsInteractions of GC-MSCs with Tumor MicroenvironmentReferenceVEGF, MIP-2, TGF-β1, IL-6, IL-8 

[23]IL-6, ET-1AKT/ERK
[26]IL-6, IL-8


JAK2/STAT3Neutrophils, macrophages[28,29]PDGF-DDβ-catenin, notch-1, NFκB, AKT
[21]VEGF, bFGF
Carcinoma-associated fibroblasts, endothelial cells[33]

## 3. MSCs Could Regulate GC Angiogenesis via Extracellular Vesicles (EVs)

GC cells release EVs in abundance. GC-EVs—which are classified in increasing order according to their size as exosomes, microvesicles and apoptotic bodies—are rich in non coding RNAs, which circulate in the circulation, saliva and urine. Conflicting data on the pro- vs. anti-angiogenic/tumoral effects of MSC-EVs in GC development are explained by the methodologic heterogeneity in EV collection, with EVs obtained from serum-deprived MSCs promoting tumor growth and EVs harvested from MSCs cultured with serum having tumor-suppressive roles. Moreover, MSC preconditioning with tumor cells and the timing of EV administration at varying GC stages can also differently influence study outcomes [34].

GC cell-derived exosomes (GC-exos) promote tumor angiogenesis by activating endothelial cells and by transforming pericytes, fibroblasts and MSCs into myofibroblasts [35].

GC exos are loaded with various cargos with the purpose of modulating angiogenesis. MiRNAs inhibit the expression of angiogenesis-related genes, but in cancer cells miRNAs are downregulated and their activity is inhibited by increased expression of other non-coding RNAs: circular RNAs (circRNAs) and long non-coding RNAs (lncRNAs). CircRNAs stimulate VEGF-A expression via the sequestration of miRNAs that directly target the 3′-untranslated region (3′-UTR) of VEGF-A, because MiRNAs downregulate VEGF-A expression in endothelial cells [36]. HUVEC tube formation has been shown to be impaired by exosomal circ29 RNA, which determines miR-29a loss of function and is highly expressed in GC [37]. By delivering miR-29a/c to GC cells using cell-derived microvesicles as carriers, angiogenesis can be inhibited in GC cells by significantly downregulating VEGFA expression [38].

Exo-miRNAs mirror the pathological changes found in the source GC cells and can be used as biomarkers of GC metastasis spreading via the hematogenous route. Exo-miRNAs have an increased expression of MiR-379-5p and miR-410-3p in serum samples of metastatic patients after stage II/III GC surgery, being able to predict metastasis before imaging confirmation [39].

Exosomes contain miRNAs that carry angiogenic molecules from tumor cells to various close or distant neovasculature related cells. Pro-angiogenic exosomes are involved in endothelial cell reprogramming by inducing the expression of angiogenic genes [13].

GC exos carrying hepatocyte growth factor small interfering RNA (HGF siRNA) have been shown to inhibit tumor growth and angiogenesis in nude mice in vivo; human embryonic kidney 293T cells were used as exosome donors. The HGF protein is upregulated and is associated with poor prognosis in GC. By silencing HGF in SGC-7901 cells with siRNA, tumor angiogenesis was also suppressed and VEGF expression was also downregulated. HUVECs co-cultured with these exosome-pretreated SGC-7901 cells exhibited less vascular ring formation [40]. As a HGF-antagonist and angiogenesis inhibitor, the NK4 gene has also been investigated in GC; human BM-MSCs have been used as a gene delivery vehicle for NK4 gene therapy for the human GC cell line MKN45. GC growth and intratumoral angiogenesis was found to be inhibited and the microvessel density was decreased—being the lowest in comparison with a lentiviral vector carrying NK4 and controls in a MSC-NK4-treated nude mouse xenograft tumor model [41].

GC exos were also loaded with the multifunctional Y-box binding protein 1 (YB-1), an element of inactive messenger RNAs that is overexpressed in the vascular endothelial cells of GC tissues and is either oncogenic or tumor-suppressive. In GC, YB-1 overexpression is associated with advanced cancer stages and increased angiogenesis. In vitro angiogenesis studies on HUVECs and SGC-7901 GC cells have shown that YB-1-overexpresseing GC exos promote increased neovasculature and tube formation as compared to negative controls and phosphate buffered saline; a lentivirus vector transfered the YBX1gene sequence into the HUVECs, leading to angiogenic effects. Pro-angiogenic factors such as VEGF, angiopoietin-1, MMP-9 and IL-8 are upregulated in HUVECs treated with YB-1-overexpressing GC exos [13].

Transplantation of human MSCs overexpressing GRP78 accelerates angiogenesis by intercellular communication. GC exos containing glucose-regulated protein 78 (GRP78) have been shown to promote angiogenesis in an in vitro study on human AGS cells. HUEhT-1 cells, an immortalized HUVEC line, were incubated with overexpressed GRP78 exos and the tube formation assay showed increased angiogenesis mediated by AKT phosphorylation [42].

## 4. MS-Derived Exosomes (MSC-Exos)

Vascular stress caused by radiation or anti-angiogenic drugs increases tumor ischemia, leading to high MSC-exo recruitment [43]. CD9 and CD63—clusters of differentiation molecules found on the cell surface—are used as biomarkers for exosomes, described as round or elliptic cell vesicles with a diameter ranging from 20 to 150 nm [8,44]. Tumor cells induce the migration of MSCs to adjacent tissues by secreting exosomes. Exosomes derived from gastric cancer cells (GC MSC-exos) have been found to affect the expression of circular RNAs in human adipose-derived MSCs [45]. Exo-miRNAs are more stable biomarkers than miRNAs [32].

MSC-exos increased human gastric carcinoma cell line HGC-27 numbers by 2–3-fold and the number of invading cells by 8-fold in an *ex vivo* study on HGC-27 cells that underwent epithelial-to-mesenchymal transition through the AKT pathway [46]. Chemotherapy stimulates the secretion of exosomes in tumor cells, spreading chemoresistant miRNAs to adjacent gastric cells [47]. MSC-exos carrying platelet-derived growth factor D (PDGFD) increase GC cell proliferation and migration in animal models. GC-MSCs produce PDGF-DD—an isoform of PDGFD—which activates the β-catenin, notch-1, NFκB and AKT signaling pathways [21]. In gastric tumors, GC-MSCs trigger the conversion of macrophages into the M2 subtype of tumor-associated macrophages (M2-like TAMs) by secreting IL-6 and IL-8 and via the activation of the JAK2/STAT3 signaling in macrophages [28].

## 5. MSC-Exos Modulate GC Angiogenesis

MSC-exos have been tested as a therapeutic vehicle for the selective delivery of genes into the tumor endothelium and microenvironment. Anti-angiogenic gene therapy strategies have used retroviral vector producers, naked DNA, adenoviral vectors and genetically engineered MSCs as angiogenesis-inhibiting gene delivery vehicles. By being naturally involved in injury repair, MSCs have the potential to differentiate into various tissues and to home in on damaged tissues. Side effects are most likely to occur when the MSCs migrate to other tissues than those needed. MSC homing efficiency has been attempted through the direct administration of MSCs into GC tissues—even if this does not guarantee the best outcomes [48]. Genetically modified stem cells have been engineered as a locally precise vehicle in tissue-specific gene therapy and the selective delivery of cancer therapeutic agents [43]. 

MSCs have also been used for gastric ulcer treatment in various animal experiments. On the third day following local injection of AD-MSCs in rats, angiogenesis was confirmed in the sutured gastric perforation—with increased levels of VEGF, cyclo-oxygenase-2 (COX-2) and TGF-β1 [49]. Due to their ability to secrete proangiogenic factors, BM-MSCs were transplanted for therapeutic purposes into the gastric wall in a rat model of acetic acid-induced gastric ulcer; BM-MSCs expressed both VEGF and hepatocyte growth factor (HGF), which stimulates gastric epithelial proliferation—but only VEGF-induced angiogenesis could be tested as a possible mechanism of rapid gastric ulcer healing [50]. 

In an in vivo mouse model of *Helicobacter pylori* infection, transplantation of BM-MSCs—which may be involved in either gastric mucosa repair or cancer—was used for inducing GC. In a cell line characterized by rapid proliferation, scirrhous gastric cancer (SGC)—which was subcutaneously engrafted together with BM-MSCs into nude mice—BM-MSCs significantly increased GC neovascularization, especially after *Helicobacter pylori* exposure; this was shown by the increased expression of CD31—a platelet-endothelial cell adhesion molecule used as a marker of angiogenesis—and the neuron-glial antigen 2 (NG2) proteoglycan, which stimulates tumor vascularization by mediating pericyte endothelial cell interactions. BM-MSCs pretreated with *Helicobacter pylori* contribute to tube formation and migration in HUVECs co-cultured with SGC cells in vitro. BM-MSC transplantation to *Helicobacter pylori*-infected GC tissues leads to overexpression of THBS4, a gene that encodes the proangiogenic thrombospondin-4 protein. Integrin α2, an adhesion receptor that responds to changes in the extracellular matrix and adjacent cells, acts as a receptor for THBS4 during endothelial cell migration. Integrin α2 shows increased levels and a proangiogenic effect in HUVECs treated with recombinant Thbs4 (rTHBS4). The THBS4/integrin α2 axis facilitates tube formation in HUVECs by activating the PI3K/AKT signaling pathway [51]. These results are shown in Table 2, revealing that MSCs support rather than suppress gastric tumor growth by promoting angiogenesis.

By injecting MSCs into the peripheral circulation of mice or rats with GC, therapeutic genes could be carried to the GC microenvironment to produce toxins that damage proliferative cells. Tumor growth may also be suppressed by inhibiting the growth of vascular endothelial cells. The differentiating potential of MSCs can be used in the tumor vasculature for developing endothelial features. BM-derived endothelial cell precursors have the advantage of being obtained from the patient themselves, posing no compatibility issues [43]. Gastric submucosa-resident MSCs (GS-MSCs) found in GC transform into endothelial cells due to the proangiogenic factors secreted by GC cells, such as VEGF and basic fibroblast growth factor (bFGF). In conditioned GC cell line media, GS-MSCs (GC-GS-MSCs) form a capillary-like structure, which is superior with respect to tube length and the number of branch points than in the conditioned human embryonic kidney 293 cell (HEK293) media that was used as a control. Tumors based on the NCI-N87 GC cell line and GS-MSCs are characterized by reduced coagulative necrosis and an increased microvascular density of cells expressing CD34 [33], which in endothelial cells have an angiogenic differentiation potential [53]. 

Human BM-MSC-exos promoted angiogenesis in co-implanted tumors induced by human SGC-7901 GC cells in an in vivo study performed in mice. The increased microvascular density and the overexpression of VEGF and the chemokine receptor CXCR4 were caused by the activation of the ERK1/2 and p38 mitogen-activated protein kinase (MAPK) pathways [52].

BM-MSCs loaded with miRNAs transduced by lentiviral vectors have been used to form exosomes which have been co-cultured with gastric cells. MiR-1228 is negatively associated with GC survival. Decreased levels of miR-1228 have been found in patients with stage III and IV GC. BM-MSC-exos carrying miR-1228 and MMP-14 have been co-cultured with SGC-7901 and MGC-823 GC cells, confirming that miR-1228 is a tumor suppressor that targets and downregulates MMP-14—a protein coding gene involved in metastasis [44].

## 6. Future Perspectives on the Use of MSC-Exos for Potential Clinical Applications and Cancer Therapeutics

The angiogenic features of GC impact the response to chemotherapy, as tumor vascular endothelial cells are more resistant than normal vascular endothelial cells to cytotoxic agents. Ramucirumab—a monoclonal antibody—and apatinib, a tyrosine kinase inhibitor, both target VEGFR2 and have been used for gastric tumor environment reprogramming. The use of various angiogenesis inhibitors may exacerbate the pro-tumor effects of tumor-associated stromal cells [54]; this makes preclinical research on MSCs all the more valuable. The GC pro- and anti-angiogenic effects of MSCs have been investigated in the context of cell line characteristics, route of administration and their ratio to tumor cells. The employed MSCs were obtained from normal tissues and were harvested in vitro in co-cultures with GC cell lines, such as MKN45, SGC-7901, MGC-823 and HGC-27; they were subsequently injected into nude mice.

MSCs seem to have the benefit of acting not only as a carrier of anti-angiogenic agents, but also as an angiogenesis modifier in their own right. So far, gastric tumors have provided solid evidence that MSCs exert pro-angiogenic activity, but the use of MSCs and MSC-exos as carriers for anti-angiogenic genes such as NK4 and TNFSF14, or anti-angiogenic miRNAs such as MiR-205-5p and miR-1228, may yield clinically useful results. The cancer cell stem niche in the gastric tumor microenvironment is supported by BM-MSCs through the upregulation of the Wnt and TGF-β signaling pathways. The diagnostic value of miRNAs in GC-MSCs is supported by the higher levels of miR-214, miR-221 and miR-222 in GC-MSCs, which may help differentiate between GC tissues and nonmalignant gastric tissues. GC-MSCs could be also used as a GC biomarker via the secretion of pro-angiogenic factors such as IL-8, given that the elevated serum levels of IL-15 derived from GC-MSCs have already been suggested as a predictor of gastric malignancy [55]. Future research on exosomal molecules derived from MSCs in GC will expand on the contribution of the GC tumor microenvironment to the therapeutic tools used in current clinical practice and will identify the angiogenic mechanisms that these highly heterogeneous gastric tumors have in common. 

## 7. Conclusions

Increased tumor vessel density and proangiogenic factor levels indicate poor prognosis in various GC cell lines. Exosomes packaged with diverse cargos upregulate the expression of these growth factors in most studies on GC angiogenesis. The exosome-mediated interactions between MSCs and tumor cells may activate or inhibit angiogenesis signaling pathways and can be used either as tumor biomarkers or for developing antiangiogenic treatment strategies.

## Figures and Tables

**Table 2 biomedicines-11-01031-t002:** Summary of the effects of MSCs and MSC-exos on GC angiogenesis.

Origin of MSCs and MSC-Exos	GC Cell Line	Cargo	Species	Effect on GC Angiogenesis	Reference
BM-MSCs	MKN45	NK4 gene	Mouse	Inhibits angiogenesis	[41]
BM-MSCs	SGC		Mouse	Promotes angiogenesis by upregulating THBS4	[51]
BM-MSC-exos	SGC-7901		Mouse	Promotes angiogenesis via the ERK1/2 and p38 MAPK pathways	[52]
BM-MSC-exos	SGC-7901, MGC-823	miR-1228	Human, in vitro	Inhibits angiogenesis by downregulating MMP-14	[44]
UC-MSC-exos	HGC-27		Human, ex vivo	Promotes angiogenesis via the Akt pathway	[46]

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
