# Peer review of "Mesenchymal Stem Cell-Derived Exosomes Modulate Angiogenesis in Gastric Cancer"

_biomedicines, 2023, doi:10.3390/biomedicines11041031_

Round 1

Reviewer 1 Report

The authors managed to compose a comprehensive and detailed literature review of a topic that perhaps does not arise the level of interest it deserves. Despite the fact that current results are limited to in-vivo or ex-vivo studies, potential clinical application for anti-angiogenic targeted therapy for cancer are massive and could exponentially increase and improve prognosis for the disease population analyzed. Specifically, this literature review collects the current body of data about the role of mesenchymal stem cells (MSC) and their different categories of extracellular vesicles (EV) as angiogenesis modulator in gastric cancer (GC) tumor microenvironment (TME).

Methods and criteria for articles selection are not disclaimed. Nonetheless, the reference list is extensive, with the most relevant points being covered, therefore allowing the reader to embrace the key concepts reached so far about the topic. BM- MSCs promote GC angiogenesis by releasing VEGFs, fibroblast-derived growth factors, platelet-derived growth factors, stromal cell derived factors-1 (SDF-1s) and cytokinesMiRNAs packaged into GC-MSC-exos are delivered to GC cells and mediate GC progression by activating endothelial cells and by transforming pericytes, fibroblasts and MSCs into myofibroblasts

MSC-exos were tested as a therapeutic vehicle for the selective delivery of genes into the tumor endothelium and microenvironment. Anti-angiogenic gene therapy strategies have used retroviral vector producers, naked DNA, adenoviral vectors and genetically engineered MSCs as angiogenesis-inhibiting gene delivery vehicles. Unfortunately, as the authors pointed out already, current clinical applicability of the results presented is limited by the characteristics of the studies, being in-vitro or in-vivo only, but the potential impact on GC prognosis that such therapies could have definitely calls for further research on anti-angiogenic targeted therapy. Gastric cancer prognosis is still poor, with most disease presentation being at late stages of disease, therefore carrying both significant surgical morbidity - for resectable disease – and medical treatment morbidity. Of course, potential side effects might appear even with novel form of therapy, these being most likely to occur when the MSC migrate to other tissues than those needed; that is also why more research is certainly needed.

Author Response

We added the literature search strategy, as follows:

A literature search was performed on Pubmed/Medline, Google Scholar and Embase with the keywords “(mesenchymal stem cells OR exosomes OR mesenchymal stem cell-derived exosomes OR MSC-Exos) AND (angiogenesis OR angiogenic OR neovascularization OR vascular endothelial growth factors) AND (gastric cancer OR stomach neoplasm)”.

Reviewer 2 Report

This review paper provided a good summary of the functions of mesenchymal stem cell-derived exosomes in the angiogenesis of gastric cancer. To understand the roles of exosomes in cancer development is a fascinating field and this review will be of interest to the readers of this journal. The reviewer suggest the publication of this manuscript after revisions:

1) Figures and schematic diagrams should be added for a better understanding.

2) More discussions should be added regarding the potential of studying mesenchymal stem cell-derived exosomes in terms of clinical applications and cancer therapeutics. 

Author Response

  1. We have added two tables. Table 1 shows how mesenchymal stem cells derived from gastric cancer tissues promote tumor angiogenesis and Table 2 describes the effects of MSCs of various origin on GC angiogenesis.

  1. The following paragraph was added:

Future perspectives on the use of MSC-exos for potential clinical application and cancer therapeutics

The angiogenic features of GC impact the response to chemotherapy, as tumor vascular endothelial cells are more resistant than normal vascular endothelial cells to cytotoxic agents. Ramucirumab, a monoclonal antibody, and apatinib, a tyrosine kinase inhibitor, are both targeting VEGFR2 and have been used for gastric tumor environ-ment reprogramming. The use of various angiogenesis inhibitors may exacerbate the pro-tumor effects of tumor-associated stromal cells. This makes the preclinical re-search of MSCs all the more valuable. GC pro- and anti-angiogenic effects of MSCs were investigated in the context of cell line characteristics, route of administration and ratio to tumor cells. The employed MSCs were obtained from normal tissues and were harvested in vitro in co-cultures with GC cell lines, such as MKN45, SGC-7901, MGC-823 and HGC‑27. They were subsequently injected to nude mice.

MSCs seem to have the benefit of acting not only as a carrier of anti-angiogenic agents, but also as an angiogenesis modifier in its own right. So far, gastric tumors have provided solid evidence that MSCs exert pro-angiogenic activity, but the use of MSCs and MSC-exos as carriers for anti-angiogenic genes, such as NK4 and TNFSF14, or an-ti-angiogenic miRNAs, such as MiR-205-5p and miR-1228, may yield clinically useful results. The cancer cell stem niche in the gastric tumor microenvironment is supported by BM-MSCs through upregulating the Wnt and TGF-β signaling pathways. The diag-nostic value of miRNAs in GC-MSCs is supported by the higher levels of miR-214, miR-221 and miR-222 in GC-MSCs, which may help differentiate between GC tissues and nonmalignant gastric tissues. GC-MSCs could be also used as a GC biomarker via secreted pro-angiogenic factors, such as IL-8, given that the elevated serum levels of IL-15 derived from GC-MSCs have already been suggested as a predictor of gastric ma-lignancy. Future research on exosomal molecules derived from MSCs in GC will ex-pand the contribution of GC tumor microenvironment to the therapeutic tools in cur-rent clinical practice and will identify the angiogenic mechanisms that the highly het-erogeneous gastric tumors have in common.